# Application of Visitor Eye Movement Information to Museum Exhibit Analysis

**Yu-Ling Hsieh [1], Ming-Feng Lee [1,\*], Guey-Shya Chen [2] and Wei-Jie Wang [2]**

[1] National Museum of Natural Science, Taichung 40453, Taiwan; ling@mail.nmns.edu.tw
[2] Institute of Educational Information and Statistics, National Taichung University of Education, Taichung 40306, Taiwan; grace@mail.ntcu.edu.tw (G.-S.C.); bms110111@gs.ntcu.edu.tw (W.-J.W.)
\* Correspondence: antonio@nmns.edu.tw

**Abstract:** The motivation of this study is that after the COVID-19 epidemic, museum exhibition visits have also been significantly affected. The purpose of this research is to better understand the visual cognition of visitors, so as to improve the application of physical field or online exhibitions. Currently, no research is available on the differences in the visitor's viewing and cognitive process with eye movements sequence analysis that stem from the exhibition planning and design of different museums. This study tracks and analyzes the eye movement trajectories of visitors and studies its relation to learning and cognition and finds the key to influencing cognition through behavioral sequence analysis of displayed content. The results show that those interested in the displayed content have better cognitive performance, are immersed in reading text, and have a substantial shift in eye movement. Contrarily, those not interested in the displayed content are distracted and often turn their attention back to the title of the content. In this study, eye movement and fixation are indicators that can be used as a reference for the future design of displays to improve the effectiveness of presenting information to a visitor. Furthermore, this research can also provide future applications in integrating the virtual world and cognitive information, in the application of AR, VR, or metaverse environment, to provide people's cognition of rapid information in the virtual environment.

**Keywords:** eye tracking; fixation; sequence analysis; eye movement

## 1. Introduction

This research is based on two main exhibitions. The first is an aboriginal-themed exhibition in a museum that shows how the aborigines experience life in the contemporary environment. Another is an exhibition on the three types of special seeds found on Lanyu Island, one of Taiwan's outlying islands, showing the characteristics of these unique species. This study explores the substantial differences in the viewing process between subjects with different layout design methods and background variables. It analyzes and explores various eye movement indicators of the subjects. The comparisons include:

(1) Differences in eye movements between people interested in the content displayed in "weak pictures and strong text";
(2) The eye movement correlation between interest in the content displayed with "equal emphasis on pictures and text" and learning performance.

### 1.1. Research Purposes

This study aims to understand the cognitive process of visitors when viewing exhibitions in museums. Understanding the correlation between visitors' attention distribution on different mediums when viewing exhibitions and their learning performance can help improve the design of exhibitions in museums. Observing external viewing behavior cannot reflect different cognitions [1]. In research design, the National Museum of Natural Science is used as the space for the research, and educational exhibition explanatory labels

are taken as an example of display content. The exhibition labels selected in this study need to cooperate with regular exhibitions in the museum. The content of these two exhibitions is mainly that they have similar pictures and text frames, but the correlation between pictures and text is different. The main language in the museum, where the Institute is located, is Taiwanese Mandarin, and it is up to the curator to decide whether or not to include English descriptions for individual exhibitions. If there are visitors who are not native Chinese speakers, they can ask foreign language guides to help. Additionally, the exhibition labels selected in this research are all designed by different designers, so the exhibition styles, exhibition picture, text, and backgrounds all have their own design concepts. The study found that visual activity will affect the visual focus of visitors' viewing, which means that it will affect people's reading focus. Therefore, the research findings are provided for designers to display the exhibition labels in a follow-up design considering improvements to visual activity. The two exhibition contents selected in the research are placed in a high-ceilinged exhibition space made of light-transmitting glass, which can be visited mainly through natural light, and has the same standard auxiliary lighting as other exhibition areas in the museum, which can provide visitors to read the exhibition labels clearly.

The theme and design style of the exhibition explanatory labels are as follows:

1. An aboriginal-themed educational exhibition: The exhibition shows how aboriginal people found their way out of the impact of the contemporary environment. In the design of the exhibition, the text tells the main story, and the pictures may not necessarily complement the text;

2. Exhibition explanatory labels on the three types of special seeds that can be found on Lanyu Island, one of Taiwan's outlying islands, show the characteristics of these unique species. In the design of the exhibition, pictures and text are equally emphasized, and pictures of seeds supplement the text. The two elements are closely related.

In the study, two different exhibition labels were selected, and the two exhibition labels were also different, but they were displayed in the same exhibition location. The size of all of the exhibition labels in the experiment was $60 \times 180$ cm (horizontal length * vertical length), and the specimens were displayed next to the exhibition labels. There was a layer of matte (pp) material on the surface of the output exhibition labels, which had a fog effect, refreshing and comfortable vision without reflection. The title text size of the aboriginal exhibition label is Microsoft JhengHei $5.4 \times 5.4$ cm, the text size is Microsoft JhengHei $1.4 \times 1.4$ cm; the title size of Taiwan's Outlying Islands—Seeds of Lanyu Island is MingLiU $2.9 \times 3.04$ cm, and the text is YuanTi $1.02 \times 1.02$ cm. All copyrights of the exhibition labels used in this research are owned by the National Museum of Natural Science of Taiwan.

The visitors' eye movement and eye movement trajectory data were recorded with the eye tracker. These data were used as indicators to study the cognition process that museum visitors undergo when viewing exhibition objects. Eye movement indicators were used for analysis to determine the attention distribution, fixation times, viewing experience, and cognitive experience of the visitors of exhibition boards of different design styles. Then, we provided suggestions for exhibition design.

The goals of this study are as follows:

1. Study the viewing differences between those interested in the displayed content and those who are not interested in the displayed content via eye movement indicators;

2. Study the relationship between the interest in displayed content and learning performance via museum visitors' eye movement data.

## 1.2. Related Research Topics

In 2020, the research on fixation and meaning construction from eye movement information in the research applied to artwork and eye movement pointed out the relationship between exhibition reading and comprehension time [2]. The 2019 Museum Eye-Tracking Study uses eye trackers to understand people's most important flaws in the exhibition research part of exhibition boards and multimedia displays, and how to apply and display eye tracker research [3].

Due to the development of various visual media, research on eye movement is more necessary for its application, such as eye movement information, heat maps, and other commonly used indicators, as well as how to use eye trackers to conduct research, providing people with many interesting and practical results [4]. There are also applications related to anatomy and physiology for which the best research on eye movement is generated. In one study, researchers sought to understand human cognitive and behavioral interaction patterns by studying the vision of the human eye [5]. Another research theme is presented in the book *Eye Tracking in User Experience Design*—namely, how to conduct research on eye tracking in social media, games, e-commerce, or special groups [6].

Visual Activity

Numerous color studies have shown that the combination of text and background colors considerably impacts readers' attention and reading [7,8]. In color psychology, visual acuity refers to the visual difference between the color of the figure and background owing to the difference in hue, brightness, saturation, as well as the area and distance between the two. In other words, the visual acuity of the color is how clearly the color can be seen. The main factors affecting visual acuity are (1) viewing distance: the closer the distance between the subject and the viewing target, the higher is the visual acuity; (2) the difference between the color and background environment: the same color viewed under different backgrounds will appear to have different levels of visual acuity. In recent years, there are also studies on the use of eye trackers in museums on the correlation between light and perception of exhibits [9].

*1.3. Eye Movement Behaviors*

1.3.1. Fixation

Fixation is an eye movement state. Usually, when reading articles or looking for objects, the fixation time is between 200 ms and 300 ms. During this time, a considerable amount of information is received, especially when viewing images. In addition, the fixation time is also related to the subject's level of concentration [10]. Fixation indicators are also used in behavioral research, such as fashion brand research [11], travel choice assessment [12], indoor signs research [13], and intersection traffic sign models [14]. The length of fixation time has different interpretations according to research questions and methods. Therefore, eye movement data needs to be integrated with other information, such as questionnaires and tests, to further infer reasonable and appropriate interpretations.

1.3.2. Fixation Location Transition

Visual behavior will show a person's inner feelings. Human beings are affected by inner cognition, and the brain nerves determine the fixation position of the eyes. The movement of the line of sight indicates the constant change in attention. The subject's attention distribution and the cognitive process can be understood by observing the sequence of eye fixation position changes through the eye tracker [15]. To understand the distribution of readers' attention, researchers have used eye trackers to observe the changes in readers' fixation positions in different time periods [16].

1.3.3. Area of Interest

The area of interest (AOI) refers to the subjects' positions to look at during the experiment. It helps us understand the distribution of attention [17,18]. The setting method and shape are different depending on the eye-tracker device. The AOI setting can be performed before or after the experiment, and the size of the area is also flexible and can be changed according to the experimental needs [19,20]. The greatest significance of setting AOI is to help researchers effectively and easily analyze eye movement data.

## 2. Materials and Methods

### 2.1. Research Targets

The targets of this research are museum visitors; the gender ratio of the two stages of this study was 52% male and 48% female, and the age range was 18 to 55 years old. In terms of the age distribution ratio, 55% were 18 to 25 years old, 25% were 25 to 35 years old, 14% were 35 to 45 years old, and 6% were over the age of 45. The subjects of the experiment were asked to participate in and watch the exhibition content for the first time, which was voluntary and free of charge. This was an individual experiment, and the subjects' visual ability was normal. The eye-tracking instrument test would pass corrective lenses before the experiment, and they were non-contact lenses. All participants participated in the study under natural light in a light-transmitting exhibition hall in the museum. The research process is discussed in two stages: (1) differences in eye movement behavior between categories regarding the aboriginal exhibition's explanatory labels; the first category is focused on the display content, whereas the second category does not deal with the display content; (2) the correlation between the degree of interest and reading comprehension regarding seed characteristics display of one of Taiwan's outlying islands, Lanyu Island. The first category comprises those who were interested in and also comprehended the content. The second category refers to those not interested in the content but comprehended it. The third category comprises those interested in the content but did not comprehend it, and the fourth category refers to those not interested in the content and did not comprehend it either. The number of subjects in each stage is shown in Table 1.

**Table 1.** Subjects for different stages.

|  | **First Group** | **Second Group** | **Third Group** | **Fourth Group** | **Total** |
|---|---|---|---|---|---|
| First Stage | 50 | 50 | 0 | 0 | 100 |
| Second Stage | 32 | 22 | 20 | 26 | 100 |

### 2.2. Research Equipment

The eye tracker used in this study was the Pupil Core glasses developed by Pupil Labs. The technical specifications of pupil core are as follows: gaze accuracy is 0.60° and precision is 0.02, sampling rate and camera solution of eye camera is 200 Hz and $192 \times 192$ px, the highest quality of world camera is 120 Hz and 480 p, and the calibration is 5 points. Additionally, the pupil capture software provided by the pupil-core eye tracker was used to collect eye movement data.

### 2.3. Research Process

In this study, the cognitive process of people visiting museums was investigated, by collecting eye movement data with head-mounted eye trackers and analyzing the exhibition labels according to the distribution of fixation positions, fixation times, fixation durations, and times relative to each area of interest (AOI). These indicators were used to analyze whether the difference in design affects the eye-tracking data of the visitors, and whether there are significant differences in the cognitive process between the categories, including whether visitors are interested in the content of the exhibition and whether they fully understand the exhibition information. In the study, the viewing time of the participants was not limited, so that the participants could read the exhibition labels to complete the purpose. Each participant's collected data of eye movement need to be larger than 0.9 confidence, which is the data field in pupil capture software. Additionally, in this research, all of the participants visited the exhibition alone. This study was designed in two parts, detailed below:

1.  We first discuss a display content with "weak pictures and strong text". A total of 100 subjects were classified according to the questionnaires after viewing the exhibition. The first group was interested in the content, and the second group was

not interested in the content. The eye movement data of the two groups were collected. Important factors that raised interest among the participants were found.

2. Using a display with "an equal emphasis on both pictures and text" as an example, we discuss the eye movement correlation between the degree of interest in the displayed content and learning performance.

### 2.4. Area of Interest (AOI) for Exhibition Explanatory Labels

In this study, we explored the differences in subjects' fixation time, frequency, and location transition between text and pictures. Seven AOI areas were set up in the aboriginal exhibition and the exhibition entitled "Outlying Islands—Seeds of Lanyu Island". Those were coded A, B, C, D, E, F, and G, where A is the title of the exhibition board; C, E, and G are pictures; B, D, and F are text, as shown in Figures 1 and 2. Figure 1 illustrates an exhibition label about the aborigines in Taiwan, in which the character is a female singer. The content of the exhibition label describes how the female singer faces the society's views on different ethnic groups as aborigines and hopes to use music to help others communicate with aboriginal people. Figure 2 is Taiwan's Outlying Islands—Seeds of Lanyu Island. The three unique seeds of Lanyu Island are listed on the display panel so that museum visitors can understand the species' characteristics and functions of these three seeds. The AOI A is Seeds of Lanyu Island, AOI B and E are description and picture of "Sterculia ceramica" seeds, AOI C and F are picture and description of "Myristica ceylanica var" and AOI D and G are description and picture of "Hernanadia nymphaeifolia".

### 2.5. The Definition of Sequential Analysis

Sequential analysis, or behavioral sequential analysis, describes the method of presenting subjects' behavior in the form of coded data. One code represents one behavior. The Z-score of the conversion value between two codes is calculated to either be or not be significant through a binomial test. A Z-score above 1.96 is significant, and the larger the value, the more evident is the conversion. The probability expectation value is calculated based on the observation samples.

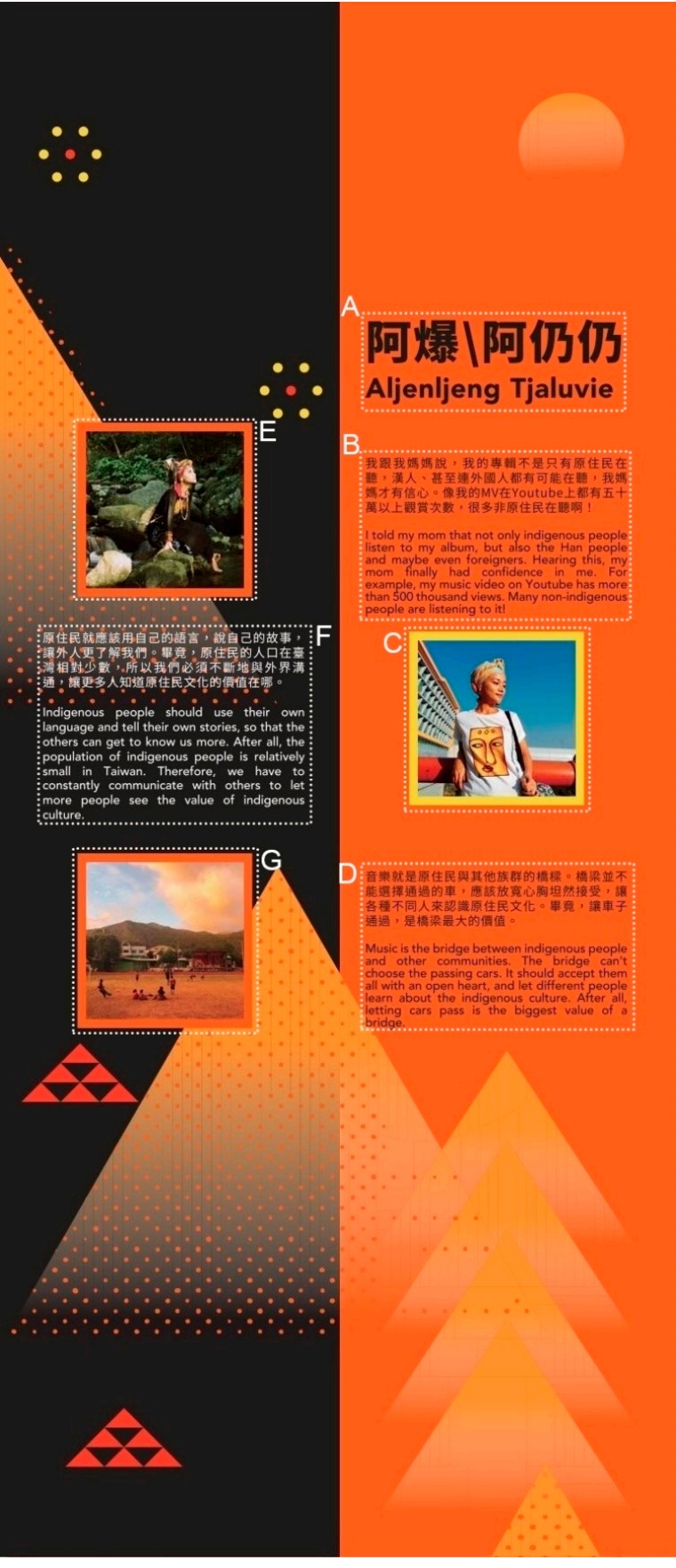

**Figure 1.** Aboriginal exhibition: comparison of AOI.

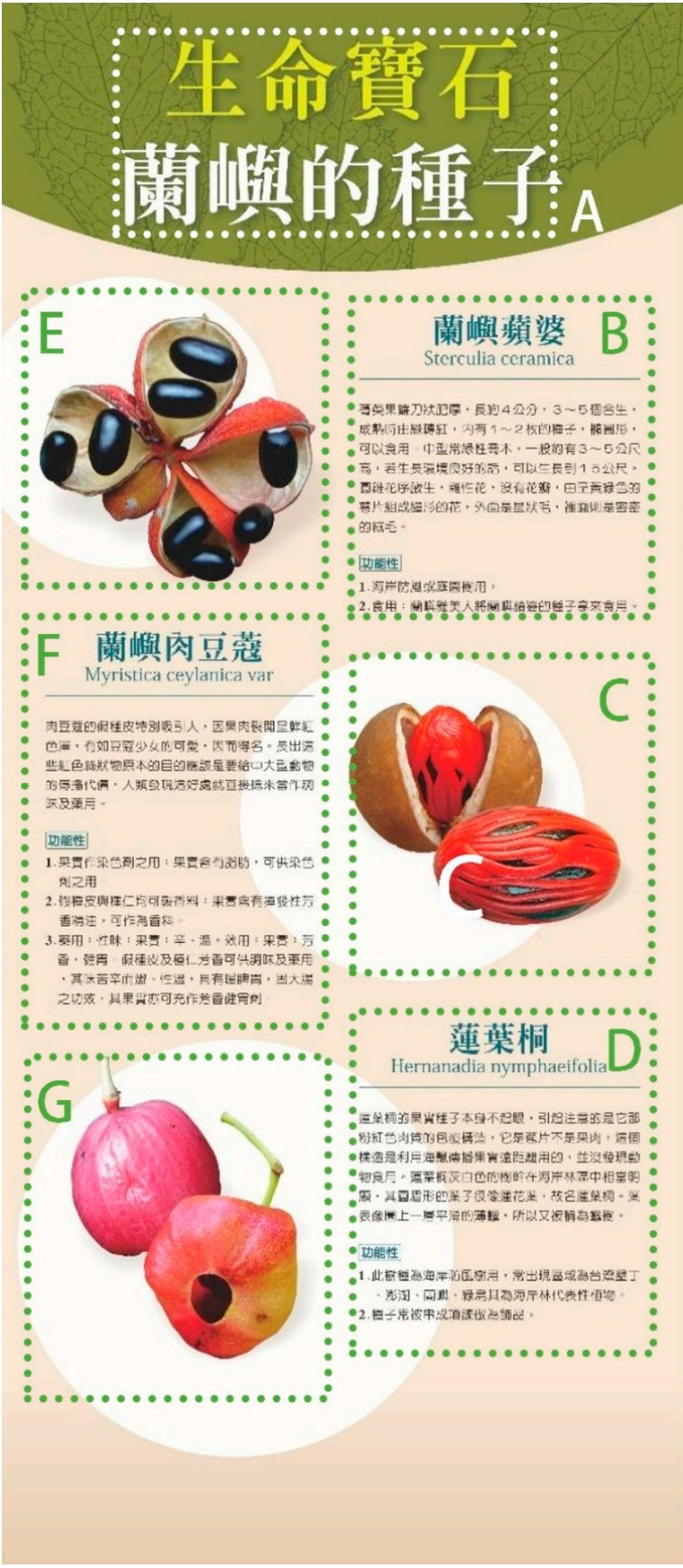

**Figure 2.** Taiwan's Outlying Islands—Seeds of Lanyu Island: comparison of AOI.

## 3. Results

### 3.1. Differences in Eye Movement between Those Interested and Not Interested in Aboriginal Exhibition Explanatory Labels

In this study, eye-tracking instruments were used to explore visitors' degree of interest in the context of an aboriginal exhibition. The results showed the differences in cognitive processes between the two groups via sequential analysis using indicators, such as AOI transitions, fixation times, and fixation percentage. The results showed considerable differences between the two groups, which are as follows:

1. The percentage of fixation times in the area of interest: Comparing the total fixation times of the two groups showed in Tables 2 and 3 that the interested group had more interest in viewing the exhibition than the less interested group. Both groups spent more than half of their time reading text, but there were considerable differences in the reading process. This is similar to the findings of Chun-Chia Wang's foreign language learning research [21]. We can observe from the percentage of the fixation time of the pictures that the AOI-C area was the highest in the two groups. According to the subjects' questionnaires, picture C was the most interesting among the displayed images. This indicates that the higher the number of fixations, the more attractive the picture is among the same type of pictures. We also confirmed the reason why picture C generated more interest by using the level of visual acuity between the picture and the background.

**Table 2.** The group interested in displayed content: AOI fixation table.

| AOI | Fixation Times | Fixation Percentage (Fixation Times of this AOI/Total Fixation Times) |
|---|---|---|
| A | 68 | 8.42% |
| B | 168 | 20.79% |
| C | 132 | 16.34% |
| D | 102 | 12.62% |
| E | 107 | 13.24% |
| F | 143 | 17.70% |
| G | 88 | 10.89% |
| Total | 808 | 100% |

**Table 3.** The group not interested in displayed content: AOI fixation table.

| AOI | Fixation Times | Fixation Percentage (Fixation Times of this AOI/Total Fixation Times) |
|---|---|---|
| A | 44 | 6.63% |
| B | 141 | 21.23% |
| C | 117 | 17.62% |
| D | 91 | 13.70% |
| E | 76 | 11.45% |
| F | 135 | 20.33% |
| G | 60 | 9.04% |
| Total | 664 | 100% |

2. Eye movement trajectory transitions: From the analysis results of the two groups of sequences, it was revealed that there were substantial differences in the reading process between the two groups in Figures 3 and 4. The AOI transfer sequences of the uninterested group increased in areas with lower visual acuity (F→E), (F→G). This indicated that it was not easy for the uninterested group to process pictures and text in areas with low visual acuity. The transition sequence of the interested group was clear, with little time spent looking back between pictures and text, and the number of AOI fixation was high. The less interested group focused on the presentation content. The results of this study are comparable to those reported in [10,22], in which eye trackers were utilized to explore changes in the general population's map reading skills and driver's driving behavior using eye movement indicators. The studies found that their eye movements increased when

the subjects were distracted or inattentive. The numerical differences were significant, and therefore, the findings could be used in the future as a reference for developing interest or distractions.

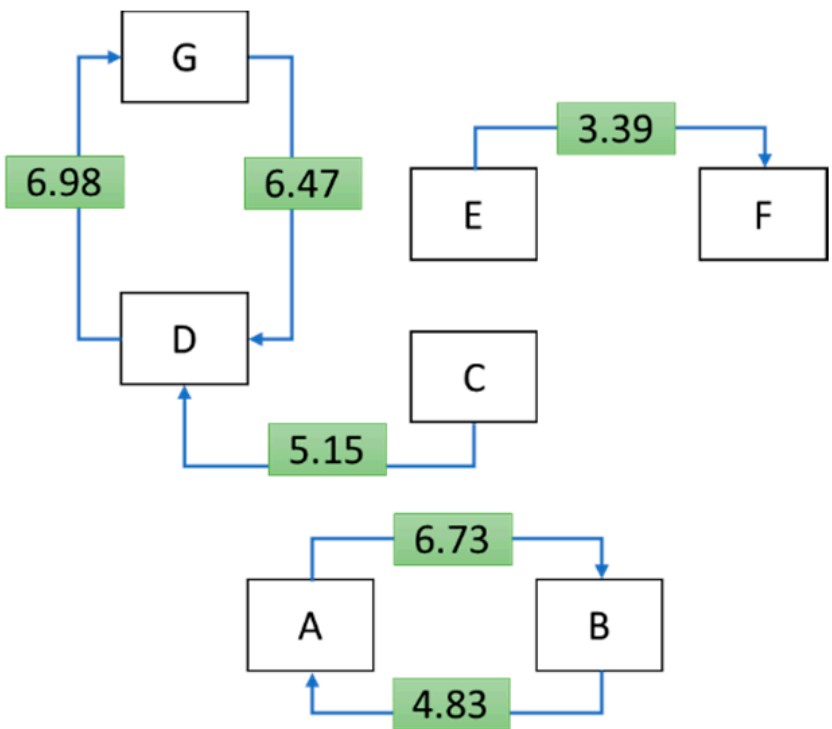

**Figure 3.** Aboriginal exhibition: sequential analysis of those interested in the displayed content.

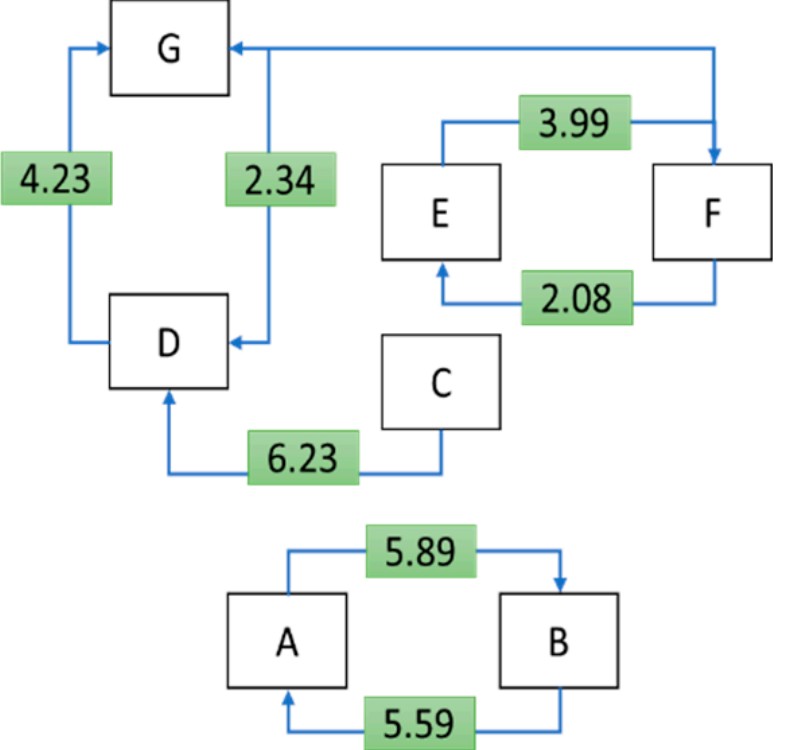

**Figure 4.** Aboriginal Exhibition: sequential analysis of those not interested in the displayed content.

In this research, according to the visual acuity index, the Y values of the YUV color coding, images, and texts in each AOI area of the exhibition were analyzed showed in Table 4. A larger value indicates that the image has a clearer contrast with the background color and is easier to notice; thus, in pictures C, E, and G, the number of fixations in the area of picture C was higher than those of pictures E and G. In Alaşhan's study, the eye movement for connected patterns and the eye movement of subjects who look at images with high contrast exhibited substantial differences [23].

**Table 4.** Aboriginal exhibition images: background color of the AOIs.

| AOI | Type | Text/Image Y Value | Background Y Value | Difference |
|-----|------|--------------------|--------------------|------------|
| C | Image | 183.528 | 110.675 | 72.853 |
| E | Image | 128.566 | 73.23 | 55.336 |
| G | Image | 105.322 | 131.881 | 26.559 |

*3.2. The Eye Movement Correlation between the Degree of Interest in the Content Displayed by the Taiwan Outlying Islands—Seeds of Lanyu Island and Learning Performance*

The study results showed that 32 of the 54 subjects, approximately 60%, who were interested in the content answered correctly. In comparison, 20 of the 46 subjects, approximately 44%, in the uninterested group answered correctly. Therefore, it can be seen from the above results that those who are interested in the content have better learning performance.

Exploring the AOI transition process between the above four groups, the four groups had some common characteristics, such as looking back and forth between AOI pictures and text (B, E), (C, F), and (D, G). These transitions were significant and indicated that all subjects from different groups could incorporate information. It was also clearly understood that the design of this scientific display is text supplemented by images. This is similar to Hayhoe's 2004 study that found a correlation between eye movement trajectory and cognition [24].

The eye movement trajectories of the four types of subjects are presented in Figures 5–8, based on which we sought to understand the factors that affect the cognitive process of reading text or processing images. The results found that the differences among the four groups showed in Tables 5–7 that were as follows: (1) The more time spent reading, the better the learning performance. Those who were more interested in the subject spent more time looking at the display. The viewing time, from the longest to the shortest, was ranked as the interested and correct group, the uninterested and correct group, the interested and incorrect group, and the uninterested and incorrect group. (2) The overall eye movement process showed that all four groups could integrate information derived from pictures and texts, but the difference in processing information can be observed from the sequence transitions on the title. Looking back less on the title meant that the interested person did not constantly look back at the title because the text content was more interesting than the title. In contrast, after reading text B or looking at picture E, those who were not interested may look back at the title because they were not interested or did not understand the knowledge that the content wants to convey, so they kept looking back and forth in the three AOIs. (3) Specific AOI transitions were substantial. The interested and correct-answer-providing group and the uninterested and correct-answer-providing group were the two groups that had better learning performance, and there were significant transitions between the text areas F and D that the other two groups did not have. These sequential transitions showed that high performers were immersed in the reading process and were interested in the text.

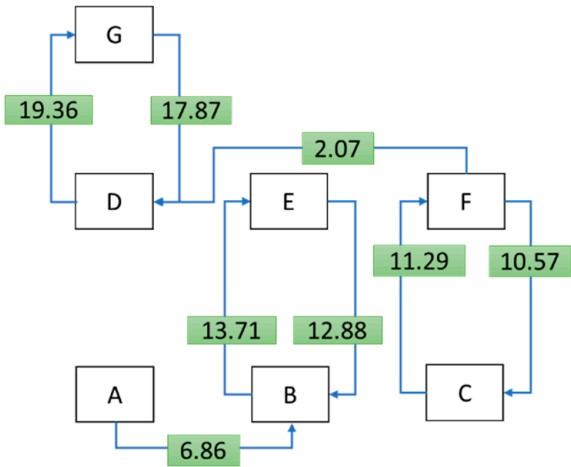

**Figure 5.** Taiwan Outlying Islands—Seeds of Lanyu Island: sequential analysis of those interested in and understood the displayed content.

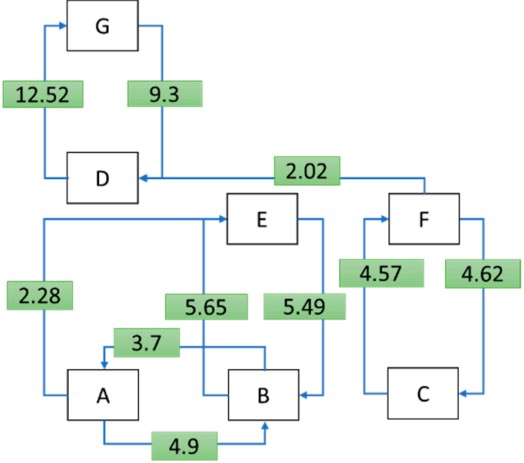

**Figure 6.** Taiwan Outlying Islands—Seeds of Lanyu Island: sequential analysis of those not interested in but who understood the displayed content.

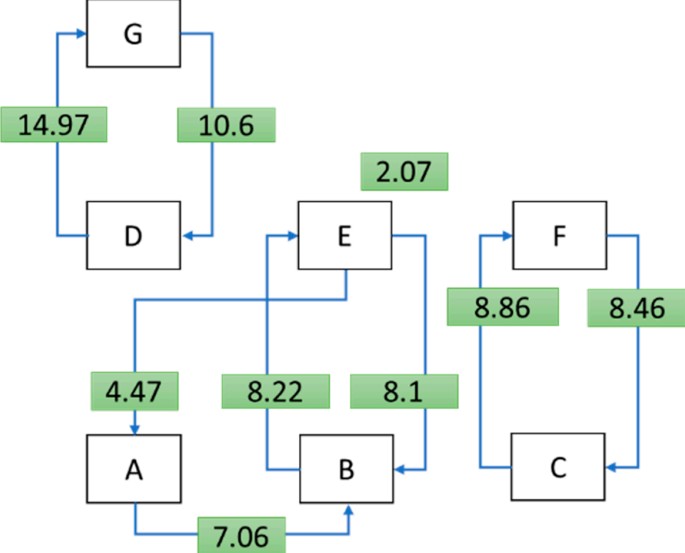

**Figure 7.** Taiwan Outlying Islands—Seeds of Lanyu Island: sequential analysis of those who were interested in but did not understand the displayed content.

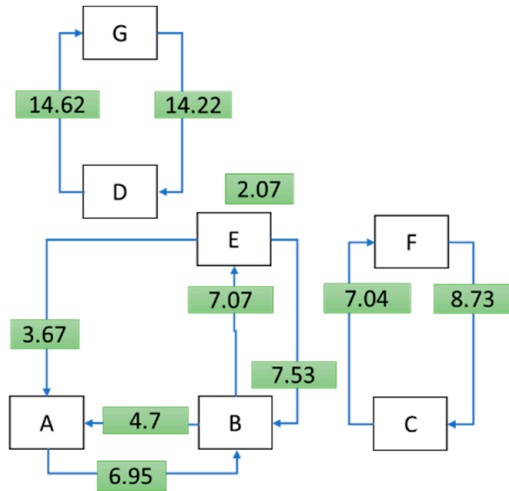

**Figure 8.** Taiwan Outlying Islands—Seeds of Lanyu Island: sequential analysis of those who were not interested in and did not understand the displayed content.

**Table 5.** The group who were interested in and understood the displayed content: AOI fixation table.

| AOI | Fixation Times | Fixation Percentage (Fixation Times of This AOI/Total Fixation Times) |
|---|---|---|
| A | 547 | 2.00% |
| B | 8143 | 29.80% |
| C | 974 | 3.60% |
| D | 7298 | 26.70% |
| E | 985 | 3.61% |
| F | 8829 | 32.28% |
| G | 548 | 2.01% |
| Total | 27,356 | 100% |

**Table 6.** The group who were not interested in but understood the displayed content: AOI fixation table.

| AOI | Fixation Times | Fixation Percentage (Fixation Times of This AOI/Total Fixation Times) |
|---|---|---|
| A | 469 | 3.08% |
| B | 4009 | 32.37% |
| C | 421 | 3.42% |
| D | 2985 | 24.10% |
| E | 517 | 4.19% |
| F | 3680 | 29.71% |
| G | 295 | 2.41% |
| Total | 12,396 | 100% |

**Table 7.** The group who were interested in but did not understand the displayed content: AOI fixation table.

| AOI | Fixation Times | Fixation Percentage (Fixation Times of This AOI/Total Fixation Times) |
|---|---|---|
| A | 668 | 5.60% |
| B | 3284 | 27.42% |
| C | 458 | 3.85% |
| D | 2847 | 23.76% |
| E | 1001 | 8.38% |
| F | 3460 | 28.87% |
| G | 252 | 2.12% |
| Total | 11,992 | 100% |

## 4. Discussion

In this research, we used eye-tracking technology to investigate the level of understanding and the cognitive process of museum visitors viewing exhibitions. Based on the research results, discussions and suggestions are put forward in this section on the following topics: "design style of exhibition explanatory labels", "prompt signals", and "museum displays".

### 4.1. Design Style of Exhibition Explanatory Labels

In this research, the eye-tracking movement was used to record the results of various indicators. Displays with high visual acuity gained more attention, had more views, and visitors did not look back to them as much. Therefore, we suggest avoiding images with low visual acuity on future exhibition explanatory labels, so the visitor can easily read the information. Furthermore, pictures and text should be given equal weight, and the color details of the pictures can reinforce the text description. This helps the visitor integrate the information between the pictures and text and further develop their cognition of the content.

### 4.2. Conspicuous Titles

Eye movement data can be used to record the information processing that occurs through reading or learning. We found the eye motion trajectory to be one of the most direct attention distribution indicators, which is similar to research findings on map reading and eye movement indicators by Weihua Dong in 2018 [17]. In this research, we used instruments to observe the eye movement behaviors of the visitor's cognitive process. For example, most visitors looked back at the title when viewing the exhibition. This behavior was the most obvious for those who were not interested in the content and had a low learning performance. Therefore, the titles of the displays should be evident, such as a one-sentence summary, allowing the visitor to better understand the core of the content. In addition, in those with a high learning performance, we observed a considerable shift in eye movement behavior while reading the text.

### 4.3. Museum Displays

Eye-tracking technology has high application value in learning and education. With the insight of visual attention and detailed eye movement data, we can objectively explain how learners process information in many aspects and, at the same time, understand the cognitive process of the learner. We suggest that future researchers add dynamic exhibition exploration by dividing the exhibition space into multiple areas of interest. Exhibition viewing behavior can then be analyzed according to eye movement indicators, exploring the differences in the learning process of visitors in dynamic exhibitions, such as human–computer interaction and videos. This information can assist display designers set up a display environment that is conducive to information processing for the general public.

### 4.4. Exhibition Content Research Limitations

The selection of exhibition content in this study was mainly based on the planning of the museum's regular exhibition and the design of the exhibition content by the curators. It is also planned that, in the future, the exhibition can be further directly planned, and experiments can be combined with the planning so that exhibitions can be designed for more experimental purposes.

### 4.5. Future Research

Over the past decade, from the gradual development of AR and VR technologies to the metaverse proposed in 2021, the application of the integration of human vision and senses has been emphasized in various research fields. The understanding of visual cognition through this research can further provide more useful information. Cognitive information

allows people to learn to recognize useful knowledge in a virtual environment, accelerate the integration of information, and further incorporate virtual information technology.

**Author Contributions:** All authors contributed to several aspects of the study. Conceptualization, Y.-L.H. and M.-F.L.; methodology, M.-F.L. and G.-S.C.; software, W.-J.W.; validation, M.-F.L. and G.-S.C.; formal analysis, M.-F.L. and G.-S.C.; investigation, M.-F.L. and Y.-L.H.; resources, Y.-L.H. and M.-F.L.; data curation, M.-F.L. and W.-J.W.; writing—original draft preparation, Y.-L.H. and M.-F.L.; writing—review and editing, M.-F.L. and G.-S.C.; supervision, M.-F.L. and G.-S.C. All authors have read and agreed to the published version of the manuscript.

**Funding:** This work was supported by the Ministry of Science and Technology, Taiwan (No. MOST109-2221-E-178-001-, No. MOST110-2410-H-178-001-).

**Institutional Review Board Statement:** Not applicable.

**Informed Consent Statement:** In this study Informed consent was obtained from all subjects involved in the study.

**Data Availability Statement:** Data are contained within the article. The data presented in this study are available in [https://drive.google.com/file/d/1ruWWD78bZbm4p_7myB0KKoO6y9qLP8i3/view?usp=sharing], (accessed on 26 May 2022).

**Conflicts of Interest:** The authors declare no conflict of interest.

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
