# Peer review of "Application of Visitor Eye Movement Information to Museum Exhibit Analysis"

_sustainability, doi:10.3390/su14116932_

Round 1

Reviewer 1 Report

Study is interesting but methodology and its description is poor:

-Title: 2x word analysis (?) Doesn't sound right.

IMPORTANT! Thanks to the description, I should be able to repeat the experiment.

-Participants: Authors should describe age, gender, nationality, recruitment method. Were participants at these exhibitions for the first time? Did they visit alone or in a group? Did they have visual impairments? Did they see colorfully? Did they have contact lenses? Were they remunerated?

-Was an initial data quality verification carried out?

-Was the visiting time measured? Were the participants restricted

-Apparatus: What ET was used ?! What kind of calibration was used 3 or 5 point? etc.

-Sofrwear: What program was used to process the data? Recommended reading: EyeTracker Data Quality: What It is and How to Measure It (Holmqvist, Nyström, and Mulvey 2012)

-Figure 1 and 2 are small, there is no care for the composition. Photos of the interiors studied are not shown. Do similar lighting conditions prevail in them? Are the height of the markings similar? Can the participants get just as close to them?

-Table 5, and 7 - why use so long numbers 0.05570380253502335?

-Where is the data availability statement?

-This is human-based research. The procedure was consulted with some institution?

-Literature connected to ET in museum - examples

Reitstätter, Luise, Hanna Brinkmann, Thiago Santini, Eva Specker, Zoya Dare, Flora Bakondi, Anna Miscená, Enkelejda Kasneci, Helmut Leder, and Raphael Rosenberg. 2020. “The Display Makes a Difference: A Mobile Eye Tracking Study on the Perception of Art Before and After a Museum’s Rearrangement.” Journal of Eye Movement Research 13.

Rusnak, Marta, and Ewa Ramus. 2019. “With an Eye Tracker at the Warsaw Rising Museum: Valorization of Adaptation of Historical Interiors.” Journal of Heritage Conservation 58: 78–90.

Tymkiw, Michael, and Tom Foulsham. 2020. “Eye Tracking, Spatial Biases and Normative Spectatorship in Museums.” Leonardo 53: 542–46.

Author Response

Response to Reviewer 1 Comments

Point 1:

-Title: 2x word analysis (?) Doesn't sound right.

IMPORTANT! Thanks to the description, I should be able to repeat the experiment.

Response 1: We removed "analysis" in the title, as it was meant to emphasize the analysis of eye movement information and the analysis of museum exhibit.

Point 2: -Participants: Authors should describe age, gender, nationality, recruitment method. Were participants at these exhibitions for the first time? Did they visit alone or in a group? Did they have visual impairments? Did they see colorfully? Did they have contact lenses? Were they remunerated?

Response 2: We have added a paragraph describing the participants' condition in Page 3.

Point 3: -Was an initial data quality verification carried out?

Response 3: Yes, the data used in our research is standard practice in the data mining method steps, all data quality has been verified, and data that is inappropriate or not completed experiments are filtered.

Point 4: -Was the visiting time measured? Were the participants restricted

Response 4: In the study, the viewing time of the participants was not limited, so that the participants could read the exhibition board to complete the purpose.

Point 5: -Apparatus: What ET was used ?! What kind of calibration was used 3 or 5 point? etc.

Response 5: The eye tracker used in this study was the Pupil Core glasses developed by Pupil Labs and used a 5-point calibration.

Point 6: -Sofrwear: What program was used to process the data? Recommended reading: EyeTracker Data Quality: What It is and How to Measure It (Holmqvist, Nyström, and Mulvey 2012)

Response 6: In the study, we used the software, pupil capture that provided by the pupil core eye tracker to collect eye movement data. And thank the reviewer for the information, which will be read and used in follow-up research.

Point 7: -Figure 1 and 2 are small, there is no care for the composition. Photos of the interiors studied are not shown. Do similar lighting conditions prevail in them? Are the height of the markings similar? Can the participants get just as close to them?

Response 7: We changed the Figure 1 and 2. In the study, the participants were carried out in natural lighting, and the experiment was carried out under the condition of being able to see the pictures and text on the exhibition board clearly. Because the participants were using head-mounted eye trackers, they were free to view the exhibition board.

Point 8: -Table 5, and 7 - why use so long numbers 0.05570380253502335?

Response 8: In the study, since all the ratios (Fixation Times of this AOI/Total Fixation Times) will be equal to 1 after adding up, in order to avoid the length of the number being subtracted from being unexplainable, the original calculation result is maintained.

Point 9: -Where is the data availability statement?

Response 9: We modified the data availability statement. The data presented in this study are available in [https://drive.google.com/file/d/1ruWWD78bZbm4p_7myB0KKoO6y9qLP8i3/view?usp=sharing]

Point 10: -This is human-based research. The procedure was consulted with some institution?

Response 10: Yes this research was pass "Human research approval".

Human research approval document[https://drive.google.com/file/d/1LqGuKD9rRM8iy8Vis2S8sLp22MEckT_V/view?usp=sharing]

Point 11: -Literature connected to ET in museum - examples

Reitstätter, Luise, Hanna Brinkmann, Thiago Santini, Eva Specker, Zoya Dare, Flora Bakondi, Anna Miscená, Enkelejda Kasneci, Helmut Leder, and Raphael Rosenberg. 2020. “The Display Makes a Difference: A Mobile Eye Tracking Study on the Perception of Art Before and After a Museum’s Rearrangement.” Journal of Eye Movement Research 13.

Rusnak, Marta, and Ewa Ramus. 2019. “With an Eye Tracker at the Warsaw Rising Museum: Valorization of Adaptation of Historical Interiors.” Journal of Heritage Conservation 58: 78–90.

Tymkiw, Michael, and Tom Foulsham. 2020. “Eye Tracking, Spatial Biases and Normative Spectatorship in Museums.” Leonardo 53: 542–46.

Response 11: Thanks the advice, Regarding the connection between eye trackers and museums, different themes have been linked to exhibition boards, museums and visual cognition, respectively, and will continue to be referenced in museums and eye tracking research.

Reviewer 2 Report

Interesting article, needs copy editing.

Round numbers

increase figure quality

include more relevant research to highlight contribution

Methods must be explained in more depth. Please cite the basic eye tracking references. Explain the process to readers who are less familiar. I have worked on several projects on eye tracking so I am familiar with the methods, but not all readers.

Pernice, K., & Nielsen, J. (2009). How to conduct eyetracking studies. Nielsen Norman Group, 945397498.

Carter, B. T., & Luke, S. G. (2020). Best practices in eye tracking research. International Journal of Psychophysiology, 155, 49-62.

Bergstrom, J. R., & Schall, A. (Eds.). (2014). Eye tracking in user experience design. Elsevier.

Bol, N., Boerman, S. C., Romano Bergstrom, J. C., & Kruikemeier, S. (2016, July). An overview of how eye tracking is used in communication research. In International conference on universal access in human-computer interaction (pp. 421-429). Springer, Cham.

Schall, A., & Bergstrom, J. R. (2014). Introduction to eye tracking. In Eye tracking in user experience design (pp. 3-26). Morgan Kaufmann.

Discussion is very common sense. Consider discussing how the findings can be used in innovative media; such as "Virtual Reality" (see X-reaIity theory with VR, "an artificial, virtual, and viewer-centered experience in which the user is enclosed in an all-encompassing 3D space that is - at least visually - sealed off from the physical environment"). Maybe in a metaverse? Or other innovative media?

Abstract is not very catchy.

Author Response

Response to Reviewer 2 Comments

Point 1:

Interesting article, needs copy editing.

 Response 1: Thank you for appreciating the point of view of this study, this manual will reconfirm the content of the manuscript.

Point 2: Round numbers

 Response 2: We modified all the numbers in tables2, 3, 5, 6 and 7.

Point 3: increase figure quality

 Response 3: we change the figure 1 and 2.

Point 4: include more relevant research to highlight contribution

Response 4: We have supplemented some literature to explain in 1.2. Related Research Topics.

Point 5: Methods must be explained in more depth. Please cite the basic eye tracking references. Explain the process to readers who are less familiar. I have worked on several projects on eye tracking so I am familiar with the methods, but not all readers.

 Response 5: We integrated the suggestions of the two reviewers and added relevant literature and content references.

Point 6: Discussion is very common sense. Consider discussing how the findings can be used in innovative media; such as "Virtual Reality" (see X-reaIity theory with VR, "an artificial, virtual, and viewer-centered experience in which the user is enclosed in an all-encompassing 3D space that is - at least visually - sealed off from the physical environment"). Maybe in a metaverse? Or other innovative media?

 Response 6: We have added a paragraph to introduce future research that can be applied to virtual technology technology and integrated cognitive behavior.

Point 7: Abstract is not very catchy

 Response 7: We add a paragraph describing the integration of this research into virtual technology technologies in the future.

Round 2

Reviewer 1 Report

Modification of illustrations and tables positively changed the whole. Now I understand it.

I do not understand why good answers to my previous questions were not included in the manuscript. This information needs to be in the text! The reader is supposed to know this.

Response 4: In the study, the viewing time of the participants were not limited, so that the participants could read the exhibition board to complete the purpose.

Response 5: The eye tracker used in this study was the Pupil Core glasses developed by Pupil Labs and used a 5-point calibration. 

Response 6: In the study, we used pupil capture software provided by the pupil core eye tracker to collect eye movement data.

Response 7:  In the study, the participants were carried out in natural lighting. 

Additionally, a lot of data is still missing to ensure the aforementioned repeatability of the experiment.

-How big were the signatures? Was the size of the boards similar? Provide description! Was the position of the boards the same (vertically, horizontally, diagonally - like on the desktop?) Were both boards matte, or were they coated with foil and shiny?

- (Connected with response 5) Please specify device accuracy, camera resolution, sampling rate, and accepted calibration parameters.

-What behaviors of the participants eliminated the collected data?  Did they visit alone or in groups?

-Maybe it's a trivial remark but it's important. The participants' descriptions must indicate what nationality they were and whether they could read in a given language?  In one museum the inscriptions are in two languages (fig.1) and in the other place in one (fig.2). This needs to be clarified and perhaps the scope of the AOI fields needs to be changed to the national language only. 

-For a future experiment! The color of the letters, the color of the background affect perception. Please read about it.

- (Connected with response 7) There was no illumination, that's little possible in museums. It is also about the setting relative to the light and whether they had a chance to shade the inscription. How to care was taken to ensure that the light conditions of reading were the same. Unless they were not to be the same and the lighting conditions were considered variable.

The description must be complete.

Description of research defects is missing. 

Author Response

Response to Reviewer 1 Comments

Point 1: I do not understand why good answers to my previous questions were not included in the manuscript. This information needs to be in the text! The reader is supposed to know this.

Response 1: We updated the answers into the manuscript.

Point 2:

-Response 4: In the study, the viewing time of the participants were not limited, so that the participants could read the exhibition board to complete the purpose.

Response 2: We added the response into the manuscript. In Page 4.

Point 3:

- Response 5: The eye tracker used in this study was the Pupil Core glasses developed by Pupil Labs and used a 5-point calibration.

Response 3: We added the subtitle ‘Research Equipment ’ into the manuscript and describe details of the eye tracker. In Page 4.

Point 4: - Response 6: In the study, we used pupil capture software provided by the pupil core eye tracker to collect eye movement data.

Response 4: We added response 6 into the manuscript. In Page 4.

Point 5: Response 7:  In the study, the participants were carried out in natural lighting.

Response 5: We added response 7 into the manuscript. In Page 4.

Point 6: Additionally, a lot of data is still missing to ensure the aforementioned repeatability of the experiment.

Response 6: In the study, we used the basic description of participant data, and used the description in paragraph 2.5 The Definition of Sequential Analysis to perform data detection and analysis of all participants. And shared the original data at (https://drive.google.com/file/d/1ruWWD78bZbm4p_7myB0KKoO6y9qLP8i3/view?usp=sharing) for repeated testing. In Page 7.

Point 7: -How big were the signatures? Was the size of the boards similar? Provide description! Was the position of the boards the same (vertically, horizontally, diagonally - like on the desktop?) Were both boards matte, or were they coated with foil and shiny?

Response 7: In the study, two different exhibition labels were selected, and the two exhibition labels were also different, but they were displayed in the same exhibition location. The size of the exhibition labels in the experiment is all 60*180 cm, and the specimens are displayed next to the exhibition labels. There is a layer of matte (pp) material on the surface of the output exhibition labels, which has a fog effect, refreshing and comfortable vision without reflection. In Page 2.

Point 8: - (Connected with response 5) Please specify device accuracy, camera resolution, sampling rate, and accepted calibration parameters.

Response 8: We added the subtitle ‘Research Equipment ’ into the manuscript and describe details of the eye tracker. In Page 4.

Point 9: -What behaviors of the participants eliminated the collected data?  Did they visit alone or in groups?

Response 9: Each participants’ collected data of eye movement need to bigger than 0.9 confidence that is the data field in pupil capture software. And all the participants visited alone in this research. In Page 4.

Point 10: -Maybe it's a trivial remark but it's important. The participants' descriptions must indicate what nationality they were and whether they could read in a given language?  In one museum the inscriptions are in two languages (fig.1) and in the other place in one (fig.2). This needs to be clarified and perhaps the scope of the AOI fields needs to be changed to the national language only.

Response 10: The display panels selected in this research need to cooperate with regular exhibitions in the museum. The content of these two exhibitions is mainly that they have similar picture and text frames, but the correlation between pictures and text is different. The main language in the museum where this institute is located is Taiwan Mandarin. Some exhibitions include English supplementary explanations, and some are not specially listed. If visitors are non-Chinese native speakers, they can ask foreign language guides for assistance. In Page.2.

Point 11: -For a future experiment! The color of the letters, the color of the background affect perception. Please read about it.

Response 11: The exhibition labels selected in this research are all designed by different designers, so the exhibition styles, exhibition picture, text and backgrounds all have their own design concepts. The study found that visual activity will affect the visual focus of visitors' viewing, which means that it will affect people's reading focus. Therefore, the research findings are provided for designers to display the exhibition labels in the follow-up design. Improvements to visual activity. Insert in Page 2.

Point 12: - (Connected with response 7) There was no illumination, that's little possible in museums. It is also about the setting relative to the light and whether they had a chance to shade the inscription. How to care was taken to ensure that the light conditions of reading were the same. Unless they were not to be the same and the lighting conditions were considered variable.

Response 12: The two exhibition contents selected in the research are placed in a high-ceilinged exhibition space made of light-transmitting glass, which can be visited mainly through natural light, and has the same standard auxiliary lighting as other exhibition areas in the museum. , which can provide visitors to read the exhibition labels clearly. Insert in Page 2.

Point 13: The description must be complete.

Response 13: We reply to all the questions and add them to the manuscript.

Point 14: Description of research defects is missing.

Response 14: The selection of exhibition content in this study is mainly based on the planning of the museum's regular exhibition and the design of the exhibition content by the curators. It is also planned that in the future, the exhibition can be further directly planned and experiments are combined with it, so that it can be designed for more experimental purposes.

We added a paragraph to describe the “Exhibition Content Research limitations”. Insert in Page 14.

Round 3

Reviewer 1 Report

Thank you for your answers. That was quick.

A few important details are still missing.

1. Were the boards hung horizontally / vertically at what height... ?

2. How can both boards be 60/180 if the proportions of Fig 1 and Fig 2 are different? This is why I asked in the previous review. I noticed an inconsistency.

3. If there are messages in two languages should be shown. The scope of AOI once includes English signatures and once not. Rather, this should be discussed. However, doesn't this increase the value of visual parameters?

4. Please describe the font tipe and size of the subtitles. 

5. Maybe photos or diagrams of the location of both of these boards shown with the immediate surroundings would be the best way out. They would dispel many doubts of readers (and mine).

6. If it was not the authors who designed these boards, I think you need to indicate the designers of the graphics. (copyright)

Best regards

Author Response

Response to Reviewer 1 Comments

Point 1: 1. Were the boards hung horizontally / vertically at what height... ?

Response 1: The size of the two exhibition panels are 60cm horizontally and 180cm vertically. In page 2.

Point 2: How can both boards be 60/180 if the proportions of Fig 1 and Fig 2 are different? This is why I asked in the previous review. I noticed an inconsistency.

Response 2: It has been replaced with the original size picture(Fig 1 and Fig 2), because there is a lot of non-content space on the top and bottom of the exhibition label, which is not within sight. In page 7, and page 10.

Point 3: If there are messages in two languages should be shown. The scope of AOI once includes English signatures and once not. Rather, this should be discussed. However, doesn't this increase the value of visual parameters?

Response 3: In this study, the relationship between pictures and text is mainly discussed, so in the AOI part, only pictures and texts are simply separated, and the Taiwan mandarin and English regions are not separated independently.

Point 4: Please describe the font tipe and size of the subtitles.

Response 4: The title text size of the aboriginal exhibition label is Microsoft JhengHei 5.4*5.4 cm, the text size is Microsoft JhengHei 1.4*1.4 cm, the title size of Taiwan's Outlying Islands - Seeds of Lanyu Island is MingLiU 2.9*3.04 cm, and the text text is YuanTi 1.02*1.02 cm. In page 2.

Point 5: Maybe photos or diagrams of the location of both of these boards shown with the immediate surroundings would be the best way out. They would dispel many doubts of readers (and mine).

Response 5: This research area is the high-ceilinged transparent glass display space of the entrance hall of the National Museum of Natural Science of Taiwan. The original exhibition panel was placed below the red arrow(*picture 1). The exhibition panel is posted on the wooden wall as a printout. At present, the two regular exhibitions have ended, and they will be explained by other exhibitions.

*picture 1 URL [https://drive.google.com/file/d/1UNQB5st3x0Zelvt0KPrNafiXy79abpNT/view?usp=sharing]

Point 6: If it was not the authors who designed these boards, I think you need to indicate the designers of the graphics. (copyright)

Response 6: All copyrights of the design files used in this research are owned by the National Museum of Natural Science of Taiwan. In page 2.